# Comparative efficacy of treatments for previously treated patients with advanced esophageal and esophagogastric junction cancer: A network meta-analysis

**Shuiyu Lin[1], Tingting Liu[2], Jun Chen[3], Guang Li[1], Jun Dang[1]***

**1** Department of Radiation Oncology, The First Hospital of China Medical University, Shenyang, China,
**2** Department of Radiation Oncology, Anshan Cancer Hospital, Anshan, China, **3** Department of Radiation Oncology, Shenyang Chest Hospital, Shenyang, China

* dangjunsy@163.com

## Abstract

### Background

It remains unclear which treatment is the most effective for previously treated patients with advanced esophageal and esophagogastric junction (EGJ) cancer. We conducted a network meta-analysis to address this important issue.

### Methods

PubMed, Embase, Cochrane Library, and Web of Science databases were searched for relevant phase II and III randomized controlled trials (RCTs). Overall survival (OS) was the primary outcome of interest, which was reported as hazard ratio (HR) and 95% confidence intervals (CIs).

### Results

Sixteen RCTs involving 3372 patients and evaluating 15 treatments were included in this network meta-analysis. Ramucirumab+chemotherapy (CT) (HR = 0.52, 95% CI: 0.35–0.77) and use of programmed death receptor 1 (PD-1) inhibitors, including camrelizumab (HR = 0.71, 95% CI: 0.57–0.88), sintilimab (HR = 0.70, 95% CI: 0.50–0.98), nivolumab (HR = 0.76, 95% CI: 0.62–0.94), and pembrolizumab (HR = 0.84, 95% CI: 0.72–0.98), conferred better OS than CT; however, this OS benefit was not observed for PD-L1 inhibitor (avelumab) and other target agents (trastuzumab, everolimus, gefitinib, and anlotinib). In subgroup analysis, ramucirumab+CT and pembrolizumab showed significant improvement in OS, when compared to CT, in esophageal/EGJ adenocarcinoma (AC) cases; moreover, all PD-1 inhibitors had significant OS advantage over CT in treating esophageal squamous cell carcinoma (SCC). Based on treatment ranking in terms of OS, ramucirumab+CT and camrelizumab were ranked the best treatments for patients with AC and SCC, respectively.

### Conclusions

Ramucirumab+CT and PD-1 inhibitors were superior to CT for previously treated cases of advanced esophageal/EGJ cancer. Ramucirumab+CT seemed to be the most effective

**Data Availability Statement:** All relevant data are within the paper and its Supporting Information files.

**Funding:** The author(s) received no specific funding for this work.

**Competing interests:** The authors have declared that no competing interests exist.

treatment in patients with esophageal/EGJ AC, while use of PD-1 inhibitors, especially cam-relizumab, was likely to be the optimal treatment in patients with esophageal SCC.

## Introduction

Esophageal cancer is characterized as an aggressive disease, and almost 50% of patients with esophageal cancer are diagnosed at an advanced stage [1]. Esophageal cancer has two main histological subtypes: esophageal adenocarcinoma (AC) and esophageal squamous cell carcinoma (SCC). Esophageal SCC is more prevalent in the upper and middle third of the esophagus, while AC usually arises from the distal third of the esophagus or the esophagogastric junction (EGJ). Systemic chemotherapy (CT) plays an essential role in the treatment of patients with advanced disease, in whom the median survival is only around 1 year [2]. In the attempts to improve the survival of this population, researchers have been investigating the efficacy of various target agents for a decade, such as those targeting epidermal growth factor receptor (EGFR) [3], vascular endothelial growth factor receptor two (VEGFR2) [4, 5], tyrosine kinase [6], HER2 gene [7, 8], and the mechanistic target of rapamycin (mTOR) pathway [9]. These agents have shown different degrees of efficacy outcomes.

More recently, there has been increased interest in immune checkpoint inhibitors (ICIs) for the treatment of advanced esophageal cancer [10–17]. Several phase III trials [12, 13, 15, 16] have demonstrated that, compared with CT, inhibitors of programmed death receptor 1 (PD-1) and its ligand, PD-L1, were associated with significant longer overall survival (OS) and a manageable safety profile in previously treated patients with advanced esophageal cancer.

Due to the lack of head-to-head comparison trials, it remains unclear whether ICIs have superior efficacies over targeted therapies; furthermore, the optimal regimen for previously treated patients with advanced esophageal/EGJ cancer remains controversial. Thus, we performed a network meta-analysis to assess the comparative efficacy and safety of different treatments, attempting to identify the most effective treatment for this patient population.

## Methods

### Literature search strategy

We conducted this network meta-analysis in accordance with the Preferred Reporting Items for Systematic Reviews and Meta-Analyses (PRISMA) guidelines (S1 Table) [18]. We systematically searched PubMed, Embase, Cochrane Library, Web of Science databases, and the recent congresses of American Society of Clinical Oncology and European Society for Medical Oncology for available studies before July 1, 2020. The search strategy is detailed in S2 Table. Manual searching of reference lists of the relevant publications were also performed.

### Inclusion and exclusion criteria

Studies were included if they met all of the following criteria: (1) phase II and III randomized controlled trials (RCTs) in recurrent or metastatic esophageal/EGJ cancer patients whose disease has progressed during or after previous systemic treatment; (2) compared ICIs or targeted therapies with CT, best supportive care (BSC), or placebo; (3) reported at least one of the following outcome data in each arm: OS, progression-free survival (PFS), objective response rate (ORR), and serious adverse events (SAEs); and (4) published in English. RCTs enrolling patients with both esophageal/EGJ cancer and gastric cancer were also included if they described the results for esophageal/EGJ cancer separately.

## Data extraction

Two investigators independently extracted the following information from each trial: first author or title of the RCT, study design, region, histological type, location, follow-up time, number of patients, interventions, hazard ratios (HRs) and their 95% confidence intervals (CIs) of PFS and OS, and odds ratios (ORs) and their 95% CIs of ORR and SAEs.

## Quality assessment

Two investigators independently assessed the risk of bias of each study using Cochrane Collaboration's tool [19], which includes the following five domains: sequence generation, allocation concealment, blinding, incomplete data, and selective reporting. A RCT was finally classified to have "low risk of bias" (all domains indicated as low risk), "high risk of bias" (one or more domains indicated as high risk), or "unclear risk of bias" (more than three domains indicated as unclear risk).

## Statistical analysis

The statistical analyses were performed by two investigators (SL and TL). The primary outcome was OS, while the secondary outcomes included PFS, ORR, and SAEs. HRs or ORs and their 95% CIs were used as summary statistics. For direct comparisons, standard pairwise meta-analysis was conducted using the Review Manager 5.3 (Cochrane Collaboration, Oxford, UK). Heterogeneity was assessed using chi-square ($\chi^2$) and I-square ($I^2$) tests. A random-effects model was used for data with $P$-value over 0.10 or $I^2$ over 50%, which indicated substantial heterogeneity; otherwise, a fixed-effects model was used.

Bayesian network meta-analyses were performed using a Markov Chain Monte Carlo simulation technique in JAGS and GeMTC package in R (https://drugis.org/software/r-packages/gemtc). As most direct evidence came from one trial, the fixed-effects consistency model was employed [20]. For each outcome measure, three Markov chains were generated automatically and run simultaneously. For each chain, 150000 sample iterations were generated with 100 000 burn-ins and a thinning interval of 10. The convergence of the model was assessed using the traces plot and Brooks-Gelman-Rubin method [21]. Surface under the cumulative ranking curve (SUCRA) method [22] was used to assess relative efficiency and safety rankings. A SUCRA of one indicates that the treatment is certain to be the best and zero if the treatment is certain to be the worst. The transitivity assumption was evaluated by comparing the distribution of potential effect modifiers (sample size, median age, and median follow-up time) across treatment comparisons [23]. Global inconsistency was assessed by comparing the fit of consistency and inconsistency models using deviance information criteria [24, 25]. Node-splitting analysis was used to assess whether there was inconsistency between direct and indirect results within the treatment loop [26], with P<0.05 indicating significant inconsistency. Sensitivity analyses were conducted to evaluate the stability of results, omitting trials with sample size of less than 50, or trials of phase II and phase II/III. In addition, we performed subgroup analyses according to histologic type. Publication bias was examined using funnel plots [27].

## Results

### Literature search results and characteristics of included RCTs

We identified 1895 records from the initial literature search and retrieved and reviewed 143 potentially eligible reports in full text (Fig 1). The relevant references were also reviewed for missed studies. Finally, 16 RCTs [3–17, 28, 29] were deemed eligible for inclusion with a total of 3372 patients enrolled to receive 15 different treatments, including PD-1/L1 inhibitors

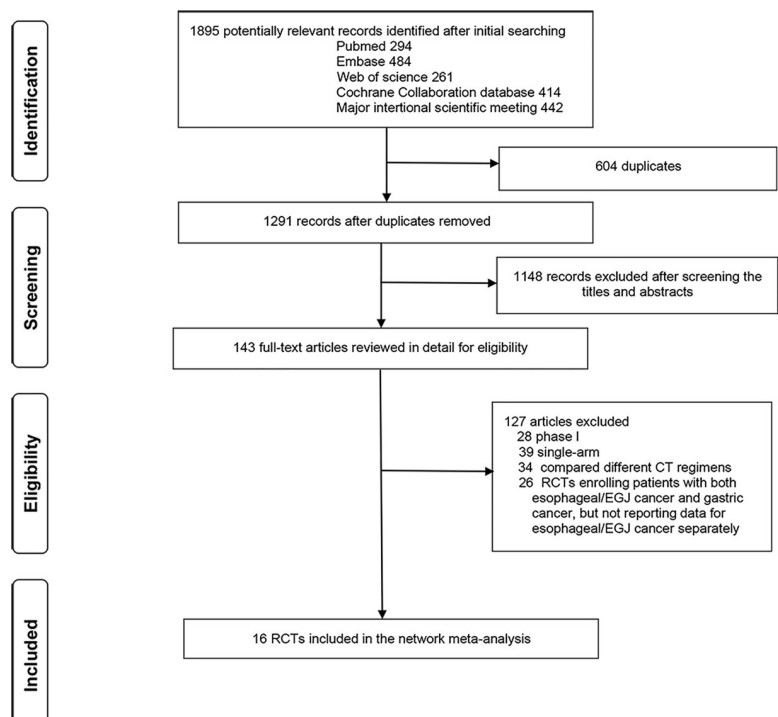

**Fig 1. Literature search and selection.** RCTs, randomized control trials; EGJ, esophagogastric junction.

(avelumab, camrelizumab, nivolumab, pembrolizumab, and sintilimab), target agents (anlotinib, everolimus, gefitinib, ramucirumab, ramucirumab+CT, trastuzumab, trastuzumab deruxtecan [T-DXd]), CT, BSC, and placebo. The baseline characteristics of the included trials are shown in Table 1. All studies were multinational trials, and a total of 12 studies (75%) were phase III trials. The median sample size was 145 participants (range, 24–628). The median age was 62 years (range, 59.1–65.5 years). The median follow-up time was 10.0 months (range, 6.7–20.7 months).

## Assessment of included trial

The risk of bias for included RCTs was summarized in S1 Fig. Overall, the risk of bias across studies was relatively low, with No RCTs rated with high risk of bias. Funnel plot analysis did not indicate any evident risk of publication bias for OS and PFS (S2 Fig).

## Conventional pairwise meta-analysis

Results of pairwise meta-analysis and individual RCTs are shown in Table 2. There were two pairwise meta-analyses for OS, including CT vs BSC or placebo and pembrolizumab vs CT. CT significantly improved OS (HR = 0.69, 95% CI: 0.51–0.92; $I^2 = 0\%$) when compared with BSC or placebo. No significant difference in OS was observed between pembrolizumab and CT (HR = 0.77, 95% CI: 0.54–1.10; $I^2 = 67\%$).

## Network meta-analysis

Fig 2 shows the network of eligible comparisons for OS and PFS. Network meta-analysis included all treatments for OS, 12 treatments for PFS, 7 treatments for ORR, and 5 treatments

**Table 1. Characteristics of included trials.**

| Trial | Design | Time Range | Region | Primary Endpoint | Treatment Details | Sample Size | Meadian Age (years) | Median Follow-up (months) | Histologic type |
|---|---|---|---|---|---|---|---|---|---|
| COG/2014 [3] | III | 2009–2011 | multicentre | OS | Gefitinib | 224 | 64.7 | NR | SCC+AC |
| | | | | | BSC/Placebo | 225 | 64.9 | | |
| RAINBOW/2014 [4] | III | 2010–2012 | multicentre | OS | Ramucirumab+CT | 66 | 61 | 7.9 | AC |
| | | | | | CT | 71 | 61 | | |
| REGARD/2014 [5] | III | 2009–2012 | multicentre | OS | Ramucirumab | 59 | 60 | NR | AC |
| | | | | | BSC/Placebo | 32 | 60 | | |
| ALTER1102/2019 [6] | II | 2016–2018 | multicentre (China) | PFS | Anlotinib | 109 | 60.6 | NR | SCC |
| | | | | | BSC/Placebo | 55 | 60.7 | | |
| GATSBY/2017 [7] | II/III | 2012–2013 | multicentre | OS | Trastuzumab | 77 | 62 | 17.5 | AC |
| | | | | | CT | 33 | 62 | 15.4 | |
| DESTINY-Gastric01/2020 [8] | II | 2017–2019 | multicentre | ORR | T-DXd | 16 | 65 | NR | AC |
| | | | | | CT | 8 | 66 | | |
| GRANITE-1/2013 [9] | III | 2009–2010 | multicentre | OS | Everolimus | 118 | 62 | 14.3 | AC |
| | | | | | BSC/Placebo | 69 | 62 | | |
| ATTRACTION-2/2017 [10] | III | 2014–2016 | multicentre (Asia) | OS | Nivolumab | 30 | 62 | 8.87 | AC |
| | | | | | BSC/Placebo | 12 | 61 | 8.59 | |
| Gastric 300/2018 [11] | III | 2015–2017 | multicentre | OS | Avelumab | 63 | 59 | 10.6 | AC |
| | | | | | CT | 48 | 61 | | |
| KEYNOTE-061/2018 [12, 13] | III | 2015–2016 | multicentre | OS/PFS | Pembrolizumab | 62 | 64 | 7.9 | AC |
| | | | | | CT | 73 | 62 | | |
| KEYNOTE-181/2019 [14] | III | NR | multicentre | OS | Pembrolizumab | 314 | 63 | 20.8 | SCC+AC |
| | | | | | CT | 314 | 62 | 20.6 | |
| Attraction-3/2019 [15] | III | 2016–2017 | multicentre (Asia) | OS | Nivolumab | 210 | 64 | 10.5 | SCC |
| | | | | | CT | 209 | 67 | 8.0 | |
| ESCORT/2019 [16] | III | 2017–2018 | multicentre (China) | OS | Camrelizumab | 228 | 60 | 8.3 | SCC |
| | | | | | CT | 220 | 60 | 6.2 | |
| ORIENT-2/2020 [17] | II | 2017–2018 | multicentre | OS | Sintilimab | 95 | 58.8 | 7.2 | SCC |
| | | | | | CT | 95 | 59.4 | 6.2 | |
| COUGAR-02/2014 [28] | III | 2008–2012 | multicentre | OS | CT | 45 | 65 | 12 | AC |
| | | | | | BSC/Placebo | 47 | 66 | | |
| TAGS/2018 [29] | III | 2016–2018 | multicentre | OS | CT | 98 | 64 | 10.7 | AC |
| | | | | | BSC/Placebo | 47 | 63 | | |

Abbreviations: OS, overall survival; PFS, progression-free survival; ORR, objective response rate; AC, adenocarcinoma; SCC, squamous cell carcinoma; T-DXd, Trastuzumab deruxtecan; CT, chemotherapy; BSC, best supportive care; NR, not reported.

for SAEs. Results of the network meta-analysis are presented in Table 3. In terms of OS (Table 3A), ramucirumab+CT, camrelizumab, sintilimab, nivolumab, and pembrolizumab were more effective than CT (HR = 0.52, 95% CI: 0.35–0.77; HR = 0.71, 95% CI: 0.57–0.88; HR = 0.70, 95% CI: 0.50–0.98; HR = 0.76, 95% CI: 0.62–0.94; and HR = 0.84, 95% CI: 0.72–0.98), gefitinib (HR = 0.39, 95% CI: 0.23–0.66; HR = 0.53, 95% CI: 0.36–0.79; HR = 0.52, 95% CI: 0.33–0.84; HR = 0.57, 95% CI: 0.39–0.84; and HR = 0.63, 95% CI: 0.43–0.91), anlotinib (HR = 0.30, 95% CI: 0.16–0.55; HR = 0.40, 95% CI: 0.24–0.68; HR = 0.40, 95% CI: 0.22–0.72; HR = 0.43, 95% CI: 0.26–0.73; and HR = 0.48, 95% CI: 0.29–0.80); ramucirumab+CT, camrelizumab, sintilimab, and nivolumab were also more effective than everolimus (HR = 0.42, 95%

**Table 2. Results of single trial and direct comparison meta-analysis.**

| Treatment | Study | OS HR(95%CI) | PFS HR(95%CI) | ORR OR(95%CI) | SAEs OR(95%CI) | Heterogeneity I² (OS) |
|---|---|---|---|---|---|---|
| Pembrolizumab vs CT | [12, 13] | 0.77(0.54–1.10) | 0.82(0.66–1.02) | 4.22(1.73–10.32) | 0.32(0.22–0.46) | 67% |
| CT vs BSC/Placebo | [28, 29] | 0.69(0.51–0.92) | 0.60(0.41–0.88) | NR | NR | 0% |
| Gefitinib vs BSC/Placebo | [3] | 0.90(0.74–1.09) | 0.80(0.66–0.96) | 6.17(0.74–51.63) | NR | |
| Ramucirumab +CT vs CT | [4] | 0.52(0.35–0.78) | 0.39(0.26–0.59) | NR | NR | |
| Ramucirumab vs BSC/Placebo | [5] | 0.76(0.47–1.21) | 0.39(0.23–0.65) | NR | NR | |
| Anlotinib vs BSC/Placebo | [6] | 1.18(0.79–1.75) | 0.46(0.32–0.66) | 2.12(0.43–10.34) | NR | |
| Trastuzumab vs CT | [7] | 1.18(0.70–2.01) | NR | NR | NR | |
| Everolimus vs BSC/Placebo | [9] | 0.84(0.61–1.16) | NR | NR | NR | |
| Nivolumab vs BSC/Placebo | [10] | 0.44(0.20–0.97) | NR | NR | NR | |
| Avelumab vs CT | [11] | 0.86(0.56–1.33) | 1.22(0.78–1.91) | 1.54(0.14–17.51) | NR | |
| Nivolumab vs CT | [14] | 0.77(0.62–0.96) | 1.08(0.87–1.34) | 0.87(0.51–1.49) | 0.13(0.08–0.20) | |
| Camrelizumab vs CT | [15] | 0.71(0.57–0.87) | 0.69(0.56–0.86) | 3.72(1.98–6.99) | 0.37(0.24–0.56) | |
| Sintilimab vs CT | [17] | 0.70(0.50–0.97) | 1.00(0.77–1.39) | 2.14(0.77–5.97) | 0.39(0.20–0.77) | |
| T-DXd vs CT | [8] | 0.68(0.21–2.15) | NR | 9.00(0.85–94.90) | NR | |

Abbreviations: OS, overall survival; PFS, progression-free survival; ORR, objective response rate; SAEs, serious adverse events; HR, hazard ratio; CI, confidence interval; OR, odds ratio; T-DXd, trastuzumab deruxtecan; CT, chemotherapy; BSC, best supportive care; NR, not reported.

CI: 0.23–0.75; HR = 0.57, 95% CI: 0.35–0.91; HR = 0.56, 95% CI: 0.33–0.96; and HR = 0.61, 95% CI: 0.38–0.97); ramucirumab+CT was also more effective than pembrolizumab (HR = 0.62, 95% CI: 0.40–0.95), ramucirumab (HR = 0.46, 95% CI: 0.23–0.91), and trastuzumab (HR = 0.44, 95% CI: 0.23–0.86).

With regard to PFS (Table 3B), ramucirumab+CT showed significant advantage over all PD-1/L1 inhibitors, including camrelizumab (HR = 0.56, 95% CI: 0.35–0.89), pembrolizumab (HR = 0.47, 95% CI: 0.30–0.75), sintilimab (HR = 0.39, 95% CI: 0.23–0.65), nivolumab (HR = 0.36, 95% CI: 0.23–0.57), and avelumab (HR = 0.32, 95% CI: 0.17–0.59); ramucirumab +CT was also more effective than other targeted therapies (except ramucirumab) and CT. Camrelizumab showed significant advantage over other PD-1/L1 inhibitors, including sintilimab (HR = 0.69, 95% CI: 0.48–0.99), nivolumab (HR = 0.64, 95% CI: 0.47–0.87), and avelumab (HR = 0.57, 95% CI: 0.35–0.93), except pembrolizumab; camrelizumab was also superior to CT and gefitinib.

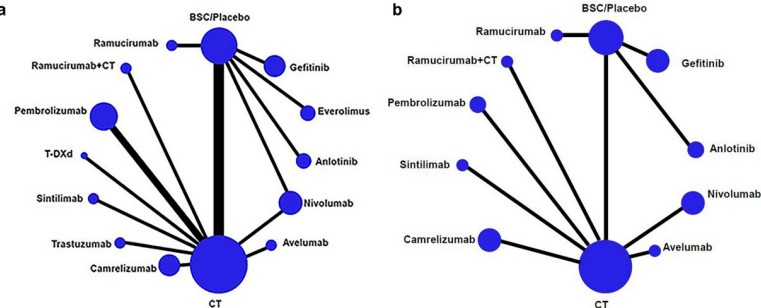

**Fig 2. Network of eligible comparisons for the network meta-analysis.** a, overall survival; b, progression-free survival. T-DXd, trastuzumab deruxtecan; CT, chemotherapy; BSC, best supportive care.

**Table 3. Results of network meta-analysis.**

**a. HR with 95%CI for OS**

| Ramucirumab+CT | | | | | | | | | | | | | |
|---|---|---|---|---|---|---|---|---|---|---|---|---|---|
| 0.73(0.47–1.15) | Camrelizumab | | | | | | | | | | | | |
| 0.74(0.45–1.25) | 1.01(0.69–1.50) | Sintilimab | | | | | | | | | | | |
| 0.68(0.44–1.07) | 0.93(0.69–1.25) | 0.92(0.62–1.37) | Nivolumab | | | | | | | | | | |
| 0.76(0.22–2.59) | 1.04(0.32–3.36) | 1.03(0.31–3.41) | 1.12(0.34–3.58) | T-DXd | | | | | | | | | |
| 0.62(0.40–0.95) | 0.84(0.65–1.09) | 0.83(0.58–1.20) | 0.90(0.69–1.18) | 0.81(0.25–2.61) | Pembrolizumab | | | | | | | | |
| 0.61(0.34–1.11) | 0.83(0.51–1.34) | 0.82(0.47–1.41) | 0.89(0.55–1.45) | 0.80(0.23–2.75) | 0.98(0.62–1.56) | Avelumab | | | | | | | |
| 0.52(0.35–0.77) | 0.71(0.57–0.88) | 0.70(0.50–0.98) | 0.76(0.62–0.94) | 0.68(0.22–2.18) | 0.84(0.72–0.98) | 0.86(0.55–1.32) | CT | | | | | | |
| 0.46(0.23–0.91) | 0.63(0.35–1.13) | 0.62(0.33–1.17) | 0.67(0.38–1.20) | 0.60(0.17–2.20) | 0.75(0.42–1.32) | 0.76(0.38–1.51) | 0.89(0.51–1.53) | Ramucirumab | | | | | |
| 0.44(0.23–0.86) | 0.60(0.34–1.06) | 0.59(0.32–1.11) | 0.65(0.36–1.13) | 0.58(0.16–2.07) | 0.71(0.41–1.23) | 0.73(0.37–1.43) | 0.85(0.50–1.43) | 0.95(0.45–2.06) | Trastuzumab | | | | |
| 0.42(0.23–0.75) | 0.57(0.35–0.91) | 0.56(0.33–0.96) | 0.61(0.38–0.97) | 0.55(0.16–1.88) | 0.67(0.43–1.06) | 0.69(0.37–1.26) | 0.80(0.52–1.23) | 0.90(0.51–1.61) | 0.94(0.48–1.87) | Everolimus | | | |
| 0.39(0.23–0.66) | 0.53(0.36–0.79) | 0.52(0.33–0.84) | 0.57(0.39–0.84) | 0.51(0.15–1.74) | 0.63(0.43–0.91) | 0.64(0.37–1.11) | 0.75(0.53–1.05) | 0.84(0.51–1.40) | 0.88(0.47–1.66) | 0.94(0.64–1.36) | Gefitinib | | |
| 0.35(0.22–0.57) | 0.48(0.34–0.68) | 0.47(0.31–0.73) | 0.51(0.37–0.72) | 0.46(0.14–1.52) | 0.57(0.41–0.78) | 0.58(0.34–0.97) | 0.67(0.51–0.89) | 0.76(0.47–1.22) | 0.79(0.44–1.45) | 0.84(0.61–1.16) | 0.90(0.74–1.09) | Placebo/BSC | |
| 0.30(0.16–0.55) | 0.40(0.24–0.68) | 0.40(0.22–0.72) | 0.43(0.26–0.73) | 0.39(0.11–1.37) | 0.48(0.29–0.80) | 0.49(0.26–0.94) | 0.57(0.35–0.92) | 0.64(0.35–1.19) | 0.67(0.33–1.38) | 0.71(0.43–1.18) | 0.76(0.49–1.18) | 0.85(0.57–1.25) | Anlotinib |

**b. HR with 95%CI for PFS**

| Ramucirumab+CT | | | | | | | | | | |
|---|---|---|---|---|---|---|---|---|---|---|
| 0.56(0.35–0.89) | Camrelizumab | | | | | | | | | |
| 0.60(0.28–1.29) | 1.06(0.54–2.10) | Ramucirumab | | | | | | | | |
| 0.51(0.26–0.99) | 0.90(0.51–1.60) | 0.85(0.45–1.60) | Anlotinib | | | | | | | |
| 0.47(0.30–0.75) | 0.84(0.62–1.14) | 0.79(0.40–1.56) | 0.94(0.53–1.64) | Pembrolizumab | | | | | | |
| 0.39(0.26–0.59) | 0.69(0.56–0.86) | 0.65(0.34–1.23) | 0.77(0.45–1.29) | 0.82(0.66–1.02) | CT | | | | | |
| 0.39(0.23–0.65) | 0.69(0.48–0.99) | 0.65(0.32–1.31) | 0.77(0.42–1.40) | 0.82(0.57–1.18) | 1.00(0.74–1.34) | Sintilimab | | | | |
| 0.36(0.23–0.57) | 0.64(0.47–0.87) | 0.60(0.31–1.18) | 0.71(0.40–1.25) | 0.76(0.56–1.03) | 0.93(0.75–1.15) | 0.93(0.64–1.33) | Nivolumab | | | |
| 0.32(0.17–0.59) | 0.57(0.35–0.93) | 0.54(0.24–1.17) | 0.63(0.32–1.25) | 0.67(0.41–1.11) | 0.82(0.53–1.29) | 0.82(0.48–1.40) | 0.88(0.54–1.46) | Avelumab | | |
| 0.29(0.16–0.53) | 0.52(0.32–0.83) | 0.49(0.28–0.85) | 0.58(0.38–0.86) | 0.61(0.38–0.99) | 0.75(0.49–1.14) | 0.75(0.45–1.25) | 0.81(0.50–1.30) | 0.91(0.49–1.69) | Gefitinib | |
| 0.23(0.13–0.41) | 0.41(0.27–0.64) | 0.39(0.23–0.66) | 0.46(0.32–0.66) | 0.49(0.32–0.76) | 0.60(0.41–0.88) | 0.60(0.37–0.97) | 0.65(0.42–1.00) | 0.73(0.41–1.32) | 0.80(0.66–0.96) | Placebo/BSC |

**c. OR with 95%CI for ORR**

| T-DXd | | | | | | |
|---|---|---|---|---|---|---|
| 2.94(0.25–97.83) | Pembrolizumab | | | | | |
| 3.39(0.33–107.26) | 1.16(0.39–3.64) | Camrelizumab | | | | |
| 5.86(0.48–198.98) | 1.99(0.48–7.97) | 1.72(0.48–5.72) | Sintilimab | | | |
| 7.22(0.13–479.39) | 2.41(0.07–37.83) | 2.09(0.07–29.27) | 1.22(0.03–19.61) | Avelumab | | |
| 12.76(1.36–387.1) | 4.38(1.85–11.66) | 3.79(2.06–7.34) | 2.21(0.81–6.77) | 1.82(0.14–54.93) | CT | |
| 14.83(1.49–461.57) | 5.04(1.83–15.32) | 4.36(1.94–10.17) | 2.54(0.82–8.87) | 2.09(0.15–66.03) | 1.15(0.67–1.98) | Nivolumab |

**d. OR with 95%CI for SAEs**

| Nivolumab | | | | |
|---|---|---|---|---|
| 0.40(0.22–0.72) | Pembrolizumab | | | |
| 0.36(0.19–0.66) | 0.88(0.50–1.55) | Camrelizumab | | |
| 0.33(0.15–0.74) | 0.82(0.39–1.76) | 0.93(0.42–2.08) | Sintilimab | |
| 0.13(0.08–0.20) | 0.32(0.22–0.46) | 0.36(0.24–0.55) | 0.39(0.20–0.75) | CT |

Abbreviations: HR, hazard ratios; OR, odds ratio; CI, confidence interval; OS, overall survival; PFS, progression-free survival; ORR, objective response rate; SAEs, serious adverse events; T-DXd, trastuzumab deruxtecan; CT, chemotherapy; BSC, best supportive care.

Regarding ORR (Table 3C), T-DXd, pembrolizumab, and camrelizumab were better than CT (OR = 12.76, 95% CI: 1.36–387.1; OR = 4.38, 95% CI: 1.85–11.66; and OR = 3.79, 95% CI: 2.06–7.34) and nivolumab (OR = 14.83, 95% CI: 1.49–461.57; OR = 5.04, 95% CI: 1.83–15.32; and OR = 4.36, 95% CI: 1.94–10.17).

In terms of SAEs (Table 3D), nivolumab was safer than pembrolizumab (OR = 0.40, 95% CI: 0.22–0.72), camrelizumab (OR = 0.36, 95% CI: 0.19–0.66), and sintilimab (OR = 0.33, 95% CI: 0.15–0.74). All PD-1 inhibitors were safer than CT. Since most of the data for treatments with target agents were extracted from esophageal/EGJ cancer subgroups of studies involving participants with both esophageal/EGJ cancer and gastric cancer in which safety profiles were not reported, we failed to obtain enough data for their SAEs. Thus, comparative safety comparisons between target agents, PD-1/L1 inhibitors, and CT could not be performed.

Results of the treatment ranking based on SUCRA are presented in Table 4, with ranking curves shown in S3 Fig. In terms of OS, ramucirumab+CT was ranked the most effective treatment (0.95), followed by camrelizumab (0.79), sintilimab (0.79), nivolumab (0.72), and T-DXd (0.70). With regard to PFS, ramucirumab+CT (0.99) and camrelizumab (0.79) were ranked the best and the second-best treatments, respectively, followed by ramucirumab (0.78), anlotinib (0.67), and pembrolizumab (0.65). As for ORR, T-DXd (0.89) was ranked the best treatment, followed by pembrolizumab (0.73), camrelizumab (0.68), sintilimab (0.48), and avelumab (0.43). In terms of SAEs, nivolumab (1.00) was the least toxic treatment, followed by pembrolizumab (0.59), camrelizumab (0.47), sintilimab (0.43), and CT (0.00).

## Transitivity, inconsistency, and sensitivity analysis

Assessment of transitivity indicated that the sample size, median age, and median follow-up time across treatments were relatively similar (S4 Fig). The fit of the consistency model was similar to that of the inconsistency model regarding all outcomes (S3 Table). There was one independent closed loop in the network for OS: nivolumab-CT-BSC/placebo. Analysis of inconsistency in OS showed that the indirect results were similar to the direct results (S5 Fig).

Sensitivity analysis omitting sample size less than 50 [8, 10], or phase II and phase II/III trials [6–8, 17] that did not affect the main results for OS (S4 Table).

## Subgroup analysis

In the subgroup analysis of esophageal/EGJ AC (11 trials with 1102 patients receiving 10 treatments) (Table 5A), ramucirumab+CT showed significant OS advantage over CT (HR = 0.52, 95% CI: 0.35–0.77), ramucirumab (HR = 0.47, 95% CI: 0.24–0.94), trastuzumab (HR = 0.44, 95% CI: 0.23–0.86), and everolimus (HR = 0.43, 95% CI: 0.24–0.77); pembrolizumab significantly improved OS when compared to CT (HR = 0.64, 95% CI: 0.45–0.91) and everolimus (HR = 0.52, 95% CI: 0.30–0.92). In terms of PFS, ramucirumab+CT was more effective than CT (HR = 0.39, 95% CI: 0.26–0.59), avelumab (HR = 0.32, 95% CI: 0.17–0.59), gefitinib (HR = 0.29, 95% CI: 0.16–0.53); ramucirumab was superior to gefitinib (HR = 0.48, 95% CI: 0.27–0.84). Based on treatment ranking (Table 4 and S3 Fig), ramucirumab+CT was ranked the most effective treatment (0.89) in terms of OS, followed by pembrolizumab (0.78) and nivolumab (0.74); ramucirumab+CT (0.98) was still the best treatment in terms of PFS, followed by ramucirumab (0.79) and CT (0.56).

In subgroup analysis of esophageal SCC (4 trials with 1458 patients receiving 5 treatments) (Table 5B), treatment with PD-1 inhibitors, including camrelizumab (HR = 0.71, 95% CI: 0.57–0.88), sintilimab (HR = 0.70, 95% CI: 0.50–0.98), nivolumab (HR = 0.77, 95% CI: 0.62–0.96), and pembrolizumab (HR = 0.78, 95% CI: 0.63–0.97), showed significant OS advantage over CT. Camrelizumab and pembrolizumab also significantly improved PFS when compared

**Table 4. Treatment ranking.**

| OS Treatment | SUCRA | PFS Treatment | SUCRA | ORR Treatment | SUCRA | SAEs Treatment | SUCRA | OS (AC) Treatment | SUCRA | OS (SCC) Treatment | SUCRA | PFS (AC) Treatment | SUCRA | PFS (SCC) Treatment | SUCRA |
|---|---|---|---|---|---|---|---|---|---|---|---|---|---|---|---|
| Ramucirumab+CT | 0.95 | Ramucirumab+CT | 0.99 | T-DXd | 0.89 | Nivolumab | 1.00 | Ramucirumab+CT | 0.89 | Camrelizumab | 0.73 | Ramucirumab+CT | 0.98 | Camrelizumab | 0.94 |
| Camrelizumab | 0.79 | Camrelizumab | 0.79 | Pembrolizumab | 0.73 | Pembrolizumab | 0.59 | Pembrolizumab | 0.78 | Sintilimab | 0.73 | Ramucirumab | 0.79 | Pembrolizumab | 0.76 |
| Sintilimab | 0.79 | Ramucirumab | 0.78 | Camrelizumab | 0.68 | Camrelizumab | 0.47 | Nivolumab | 0.74 | Nivolumab | 0.53 | CT | 0.56 | Sintilimab | 0.32 |
| Nivolumab | 0.72 | Anlotinib | 0.67 | Sintilimab | 0.48 | Sintilimab | 0.43 | T-DXd | 0.66 | Pembrolizumab | 0.50 | Avelumab | 0.35 | CT | 0.32 |
| T-DXd | 0.70 | Pembrolizumab | 0.65 | Avelumab | 0.43 | CT | 0.00 | Avelumab | 0.56 | CT | 0.01 | Gefitinib | 0.29 | Nivolumab | 0.15 |
| Pembrolizumab | 0.62 | CT | 0.43 | CT | 0.18 |  |  | CT | 0.43 |  |  | Placebo/BSC | 0.04 |  |  |
| Avelumab | 0.60 | Sintilimab | 0.42 | Nivolumab | 0.11 |  |  | Ramucirumab | 0.34 |  |  |  |  |  |  |
| CT | 0.44 | Nivolumab | 0.33 |  |  |  |  | Trastuzumab | 0.28 |  |  |  |  |  |  |
| Ramucirumab | 0.38 | Avelumab | 0.24 |  |  |  |  | Everolimus | 0.25 |  |  |  |  |  |  |
| Trastuzumab | 0.33 | Gefitinib | 0.18 |  |  |  |  |  |  |  |  |  |  |  |  |
| Everolimus | 0.29 | Placebo/BSC | 0.02 |  |  |  |  |  |  |  |  |  |  |  |  |
| Gefitinib | 0.23 |  |  |  |  |  |  |  |  |  |  |  |  |  |  |
| Placebo/BSC | 0.12 |  |  |  |  |  |  |  |  |  |  |  |  |  |  |
| Anlotinib | 0.06 |  |  |  |  |  |  |  |  |  |  |  |  |  |  |

Abbreviations: SUCRA, surface under the cumulative ranking; OS, overall survival; PFS, progression-free survival; ORR, objective response rate; SAEs, serious adverse events; AC, adenocarcinoma; SCC, squamous cell carcinoma; T-DXd, trastuzumab deruxtecan; CT, chemotherapy; BSC, best supportive care.

**Table 5. Indirect results of subgroup analysis according to histological type.**

**a. AC**

**HR with 95%CI for OS**

| Ramucirumab+CT | Pembrolizumab | Nivolumab | T-DXd | Aelumab | CT | Ramucirumab | Trastuzumab | Everolimus | Placebo/BSC |
|---|---|---|---|---|---|---|---|---|---|
| 0.81(0.48–1.38) | Pembrolizumab | | | | | | | | |
| 0.81(0.32–2.09) | 1.00(0.40–2.51) | Nivolumab | | | | | | | |
| 0.76(0.22–2.60) | 0.94(0.28–3.20) | 0.93(0.22–3.96) | T-DXd | | | | | | |
| 0.61(0.34–1.08) | 0.75(0.43–1.30) | 0.74(0.29–1.90) | 0.79(0.23–2.72) | Aelumab | | | | | |
| **0.52(0.35–0.77)** | **0.64(0.45–0.91)** | 0.64(0.27–1.49) | 0.68(0.21–2.19) | 0.86(0.56–1.32) | CT | | | | |
| **0.47(0.24–0.94)** | 0.58(0.30–1.12) | 0.58(0.23–1.45) | 0.61(0.17–2.25) | 0.78(0.38–1.58) | 0.90(0.52–1.58) | Ramucirumab | | | |
| **0.44(0.23–0.86)** | 0.54(0.28–1.03) | 0.54(0.20–1.48) | 0.58(0.16–2.11) | 0.73(0.37–1.43) | 0.85(0.50–1.44) | 0.94(0.43–2.03) | Trastuzumab | | |
| **0.43(0.24–0.77)** | **0.52(0.30–0.92)** | 0.52(0.22–1.22) | 0.56(0.16–1.95) | 0.70(0.38–1.30) | 0.82(0.53–1.27) | 0.91(0.51–1.60) | 0.97(0.49–1.93) | Everolimus | |
| **0.36(0.22–0.59)** | **0.44(0.28–0.70)** | **0.44(0.20–0.96)** | 0.47(0.14–1.56) | 0.59(0.35–1.00) | **0.69(0.51–0.92)** | 0.76(0.47–1.22) | 0.81(0.44–1.45) | 0.84(0.61–1.15) | Placebo/BSC |

**HR with 95%CI for PFS**

| Ramucirumab+CT | Ramucirumab | CT | Avelumab | Gefitinib | Placebo/BSC |
|---|---|---|---|---|---|
| 0.60(0.28–1.29) | Ramucirumab | | | | |
| **0.39(0.26–0.59)** | 0.65(0.34–1.24) | CT | | | |
| **0.32(0.17–0.59)** | 0.53(0.24–1.17) | 0.82(0.52–1.29) | Avelumab | | |
| **0.29(0.16–0.53)** | **0.48(0.27–0.84)** | 0.74(0.48–1.15) | 0.90(0.48–1.70) | Gefitinib | |
| **0.23(0.13–0.41)** | **0.39(0.23–0.66)** | **0.60(0.41–0.88)** | 0.73(0.40–1.32) | 0.81(0.65–1.01) | Placebo/BSC |

**b. SCC**

**HR with 95%CI for OS**

| Camrelizumab | Sintilimab | Nivolumab | Pembrolizumab | CT |
|---|---|---|---|---|
| 1.01(0.68–1.50) | Sintilimab | | | |
| 0.92(0.68–1.24) | 0.91(0.61–1.35) | Nivolumab | | |
| 0.91(0.67–1.23) | 0.90(0.61–1.33) | 0.99(0.73–1.33) | Pembrolizumab | |
| **0.71(0.57–0.88)** | **0.70(0.50–0.98)** | **0.77(0.62–0.96)** | **0.78(0.63–0.97)** | CT |

**HR with 95%CI for PFS**

| Camrelizumab | Pembrolizumab | Sintilimab | CT | Nivolumab |
|---|---|---|---|---|
| 0.87(0.64–1.19) | Pembrolizumab | | | |
| 0.69(0.48–1.00) | 0.79(0.54–1.15) | Sintilimab | | |
| **0.69(0.56–0.85)** | **0.79(0.63–0.99)** | 1.00(0.74–1.34) | CT | |
| **0.64(0.47–0.87)** | 0.73(0.54–1.00) | 0.93(0.64–1.33) | 0.93(0.75–1.15) | Nivolumab |

Abbreviations: HR, hazard ratios; CI, confidence interval; OS, overall survival; PFS, progression-free survival; T-DXd, trastuzumab deruxtecan; CT, chemotherapy; BSC, best supportive care; AC, adenocarcinoma; SCC, squamous cell carcinoma.

to CT (HR = 0.69, 95% CI: 0.56–0.85; and HR = 0.79, 95% CI: 0.63–0.99); camrelizumab also had significant PFS advantage over nivolumab (HR = 0.64, 95% CI: 0.47–0.87). According to treatment ranking (Table 4 and S3 Fig), camrelizumab was ranked the most effective treatment (0.73) in terms of OS, followed by sintilimab (0.73) and nivolumab (0.53); camrelizumab (0.94) remained the best treatment in terms of PFS, followed by pembrolizumab (0.78) and sintilimab (0.32).

## Discussion

To the best of our knowledge, this is the first network meta-analysis that assessed the comparative efficacy of major treatments for previously treated patients with advanced esophageal/EGJ cancer. Our network meta-analysis showed that ramucirumab + CT and PD-1 inhibitors (camrelizumab, sintilimab, nivolumab, and pembrolizumab) conferred better OS than CT, while an OS benefit was not observed for PD-L1 inhibitor (avelumab) and other target agents (trastuzumab, everolimus, gefitinib, and anlotinib).

It should be noted that esophageal AC and SCC are generally considered to be two completely different diseases, with different molecular profiles, with distal esophageal AC showing almost the same molecular profile as junction AC [30]. In the RAINBOW trial [4], the addition of ramucirumab to CT significantly increased OS (HR = 0.52, 95% CI: 0.35–0.78) in patients with advanced EGJ AC. However, this regimen has not been tested in patients with SCC yet. More recently, several phase III trials have assessed the efficacy of PD-1/L1 inhibitors as second-line therapy in advanced esophageal or EGJ AC, but with inconsistent results. For example, pembrolizumab did not significantly improved OS, relative to CT, for patients with advanced EGJ AC in KEYNOTE-181 trial [14], but showed a positive result in KEYNOTE-061 study [12, 13]. Based on treatment ranking in terms of both OS and PFS in our network meta-analysis, ramucirumab + CT was ranked the best treatment in patients with esophageal or EGJ AC; PD-1/L1 inhibitors (pembrolizumab, nivolumab, and avelumab) were less effective than ramucirumab + CT. Conversely, PD-1 inhibitors, including camrelizumab [16], pembrolizumab [14], nivolumab [15], and sintilimab [17], have shown consistently significant longer OS than CT in previously treated patients with advanced esophageal SCC. In our network meta-analysis, despite the fact that no significant difference in OS was observed between these PD-1 inhibitors, camrelizumab was ranked the most effective treatment, either in OS or in PFS. These findings will be helpful for physicians to select more suitable therapy strategy in patients with different histological types.

Although PD-1 inhibitors have shown promising results in treatment of advanced esophageal SCC, they were likely to be more effective in patients with high PD-L1 levels. In the KEYNOTE-181 [14], pembrolizumab significantly improved OS vs CT as second-line therapy only for SCC patients with PD-L1 CPS ≥10. In ATTRACTION-3 [15], and ESCORT trials [16], despite nivolumab and camrelizumab showing OS advantage over CT, regardless of PD-L1 expression, patients with a high PD-L1 expression (CPS≥10 or PD-L1≥1) benefited more from PD-1 inhibitors. Predictive role of PD-L1 expression was also evaluated in patients with advanced esophageal/EGJ AC, but with inconsistent results. ATTRACTION-2 [10] and JAVELIN Gastric 300 [11] trials did not show a strong link between efficacy of nivolumab/avelumab and tumor PD-L1 level; meanwhile, long-term analysis of KEYNOTE-061 trial found that second-line pembrolizumab prolonged OS only among patients with PD-L1-positive esophageal/EGJ AC [12, 13]. Thus, PD-L1 expression, when used as a predictive biomarker for esophageal/EGJ AC, needs further evaluation.

Recently, the use of PD-1/L1 inhibitors, in combination with CT or CTLA-4 inhibitors, has demonstrated survival advantage over monotherapy in several tumors [31–34]. However,

these combinations have never been assessed in advanced esophageal cancer. In the present meta-analysis, PD-1/L1 inhibitors, including nivolumab and avelumab, did not show a significant OS advantage over any of the treatments (except BSC/placebo) in patients with esophageal/EGJ AC; pembrolizumab significantly improved OS when compared to CT and some target agents, but with lesser efficacy than ramucirumab+CT. For esophageal SCC, although each PD-1 inhibitor monotherapy was superior to CT in individual trials, the difference was not significant for patients with low PD-L1 expression. Thus, there is a need for large phase III trials to assess whether PD-1/L1 inhibitors + CT could significantly improve survival when compared to monotherapy, especially for patients with advanced esophageal/EGJ AC and those with low PD-L1 expression.

Based on current findings, ramucirumab+CT and camrelizumab appeared to be the best second-line treatment for patients with esophageal/EGJ AC and esophageal SCC, respectively. However, this network meta-analysis has some limitations. First, the meta-analysis was conducted based on the results reported from trials rather than individual patient data, and on indirect comparisons instead of direct comparisons. In addition, PD-1 inhibitors are likely more effective for patients with SCC and tumors with high PD-L1 expression. For those with negative or low PD-L1 level tumors, whether PD-1 inhibitors are still superior to other treatments remain uncertain. Since all studies of targeted therapies did not report the PD-L1 expression level of the patients, we could not further assess the comparative efficacy according to PD-L1 expression status. Moreover, SAEs data for ramucirumab + CT was not provided in individual trials, and thus, we could not investigate the comparative safety profile of this treatment. All the limitations mentioned above do not allow us to reach a definitive conclusion about which was the best treatment, and our findings should be interpreted with caution. Second, different CT regimens and schedules used in individual trials were grouped together, which might lead to heterogeneity. Third, some of the newer data were extracted from recent conference abstracts [6, 8, 14, 17]. This could lead to a selection bias because more survival data might be reported in the full publication. Finally, most of the data for target agents were extracted from esophageal/EGJ cancer subgroups of studies involving participants with both esophageal/EGJ cancer and gastric cancer, which may result in bias.

## Conclusions

Ramucirumab+CT and PD-1 inhibitors were superior to CT for previously treated advanced esophageal/EGJ cancer. Ramucirumab+CT seemed to be the most effective treatment in patients with esophageal/EGJ AC; moreover, PD-1 inhibitors, especially camrelizumab, were likely to be the optimal selection of treatments in patients with esophageal SCC. Future head-to-head comparison trials are needed to confirm these findings. There is also a need for phase III trials focusing on PD-1/L1 inhibitor-based combination therapy and treatment strategies in esophageal cancer patients with negative or low PD-L1 level tumors.

## Supporting information

**S1 Fig. Assessment of risk of bias.** a: Methodological quality graph: authors' judgment about each methodological quality item presented as percentages across all included studies; b: Methodological quality summary: authors' judgment about each methodological quality item for each included study, "+" low risk of bias; "?" unclear risk of bias; "-" high risk of bias. (DOC)

**S2 Fig. Comparison-adjusted funnel plots of publication bias.** a, overall survival; b, progression-free survival. T-DXd, trastuzumab deruxtecan; CT, chemotherapy; BSC, best supportive

care.
(DOC)

**S3 Fig. Treatment ranking curves based on SUCRA.** OS, overall survival; PFS, progression-free survival; ORR, objective response rate; SAE, serious adverse event; SUCRA, surface under the cumulative ranking; T-DXd, trastuzumab deruxtecan; CT, chemotherapy; BSC, best supportive care.
(DOC)

**S4 Fig. Assessment of transitivity among included trials.** a, sample size; b, median age; c, median follow up time. T-DXd, trastuzumab deruxtecan; CT, chemotherapy; BSC, best supportive care.
(DOC)

**S5 Fig. Inconsistency evaluation by node-splitting analysis for overall survival.** CT, chemotherapy; BSC, best supportive care.
(DOC)

**S1 Table. PRISMA checklist.**
(DOC)

**S2 Table. Search strategy.**
(DOC)

**S3 Table. Comparisons of the fit of consistency and inconsistency models.**
(DOC)

**S4 Table. Results of sensitivity analysis.**
(DOC)

## Author Contributions

**Conceptualization:** Jun Dang.

**Data curation:** Shuiyu Lin, Tingting Liu.

**Formal analysis:** Jun Chen, Guang Li, Jun Dang.

**Methodology:** Shuiyu Lin, Tingting Liu.

**Software:** Shuiyu Lin, Tingting Liu, Jun Chen.

**Writing – original draft:** Shuiyu Lin, Tingting Liu, Jun Chen, Guang Li.

**Writing – review & editing:** Jun Dang.

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
