## [Decision Letter · Decision Letter 0]

30 Mar 2021

PONE-D-20-24414

Comparative efficiency of treatments for previously treated patients with advanced esophageal and esophagogastric junction cancer: a network meta-analysis

PLOS ONE

Dear Dr. Dang,

Thank you for submitting your manuscript to PLOS ONE. After careful consideration, we feel that it has merit but does not fully meet PLOS ONE’s publication criteria as it currently stands. Therefore, we invite you to submit a revised version of the manuscript that addresses the points raised during the review process.

1- Please, follow the PRISMA statement when reporting the manuscript.

2- Language needs improvement and revision. For example, the title should read "efficacy" not "efficiency". The term efficiency makes more mechanical sense and is usually related to instruments while efficacy usually refers to the positive impact or the effect of a treatment on patient outcomes. Please, revise all language.

3- Address the reviewers comments below.

After revision, the article will undergo another cycle of peer-review.

We look forward to receiving your revised manuscript.

Kind regards,

Ahmed Negida, MD

Academic Editor

PLOS ONE

Journal Requirements:

Reviewers' comments:

Reviewer's Responses to Questions

**Comments to the Author**

1. Is the manuscript technically sound, and do the data support the conclusions?

Reviewer #1: No

Reviewer #2: Yes

Reviewer #3: Yes

Reviewer #4: Yes

2. Has the statistical analysis been performed appropriately and rigorously? 

Reviewer #1: I Don't Know

Reviewer #2: N/A

Reviewer #3: No

Reviewer #4: I Don't Know

3. Have the authors made all data underlying the findings in their manuscript fully available?

Reviewer #1: Yes

Reviewer #2: Yes

Reviewer #3: Yes

Reviewer #4: No

4. Is the manuscript presented in an intelligible fashion and written in standard English?

Reviewer #1: No

Reviewer #2: Yes

Reviewer #3: Yes

Reviewer #4: Yes

5. Review Comments to the Author

Reviewer #1: The authors need to clarify the investigator(s) who actually performed the statistical analyses.

The authors need to further expand on the challenges associated with data from abstract presentations only in this type of analysis.

Reviewer #2: The research method is appropriate and the research process is complete, but the research innovation is insufficient, so the selection of articles needs to be further optimized. The depth of research is not enough, and further analysis is needed.

Reviewer #3: In this manuscript, the authors made a network meta-analysis about comparative efficiency of treatments for previously treated patients with advanced esophageal and esophagogastric junction cancer.

Major revision:

1. There was no quality assessment of the included studies;

2. The charts of sensitivity analysis and heterogeneity analysis were not provided.

Reviewer #4: Based on the systematic paper review, the authors conducted a network meta-analysis to investigate which the most effective treatment for previously treated patients with advanced esophageal and esophagogastic junction cancer. They found that Ramucirumab+CT and PD-1 inhibitors were superior to CT for previously treated advanced esophageal/EGJ cancer. I think the data is valuable. They also discuss the limitation of the current study and pointing out future directions to confirm the results found in the current study.

Page 11, the first time use of network meta-analysis need to be the full name other that the abbreviation.

6. PLOS authors have the option to publish the peer review history of their article (what does this mean?). If published, this will include your full peer review and any attached files.

Reviewer #1: No

Reviewer #2: No

Reviewer #3: No

Reviewer #4: No

---

## [Author Response · Author response to Decision Letter 0]

8 Apr 2021

5. Review Comments to the Author

Reviewer #1: The authors need to clarify the investigator(s) who actually performed the statistical analyses.

The authors need to further expand on the challenges associated with data from abstract presentations only in this type of analysis.

Response: The statistical analyses were performed by two investigators (SL and TL), and we have added the sentence to the manuscript (page, line). According to the reviewer’s request, we have further discussed limitations of our findings in the last paragraph of Discussion section, as below:

Based on current findings, Ramucirumab+CT and camrelizumab appeared to be the best second-line treatment for patients with esophageal/EGJ AC and esophageal SCC, respectively. However, this network meta-analysis has some limitations. First, the meta-analysis was conducted based on the results reported from trials rather than individual patient data, and on indirect comparisons instead of direct comparisons. In addition, PD-1 inhibitors are likely more effective for patients with SCC and tumors with high PD-L1 expression. For those with negative or low PD-L1 level tumors, whether PD-1 inhibitors are still superior to other treatments remain uncertain. Since all studies of targeted therapies did not report the PD-L1 expression level of the patients, we could not further assess the comparative efficacy according to PD-L1 expression status. Moreover, SAEs data for Ramucirumab + CT was not provided in individual trials, and thus, we could not investigate the comparative safety profile of this treatment. All the limitations mentioned above do not allow us to reach a definitive conclusion about which was the best treatment, and our findings should be interpreted with caution. 

We have tried to present our findings in more probabilistic terms, for example, using words of “seem to be”, or “is likely to be”, etc.

Reviewer #2: The research method is appropriate and the research process is complete, but the research innovation is insufficient, so the selection of articles needs to be further optimized. The depth of research is not enough, and further analysis is needed.

Response: With regard to the research innovation of our study, perhaps not be sufficient enough just like the reviewer pointed out, we would like to present some points as below. For patients with esophageal squamous cell carcinoma (SCC), four phase III trials have consistently demonstrated that PD-1inhibitors (including camrelizumab, sintilimab, nivolumab, and pembrolizumab) are superior to CT. However, whether there are difference in efficacy between the four PD-1inhibitors and which might be a better selection for this set of patients remain uncertain. For patients with esophageal or EGJ adenocarcinoma (AC), there are also four phase III trials assessing efficacy of PD-1/PD-L1 inhibitors, but with inconsistent results. Whether PD-1/PD-L1 inhibitors are more effective than other targeted therapies esophageal or EGJ AC is also unclear. In light of these issues, and due to lack of head-to-head comparison trials, we performed the network meta-analysis, attempting to identify the most preferable treatment for patients with esophageal/EGJ AC and esophageal SCC, respectively. To our knowledge, this is the first network meta-analysis focusing on this subject. We found that Ramucirumab+CT seemed to be the most effective treatment in patients with esophageal/EGJ AC; while PD-1 inhibitors, especially camrelizumab, were likely to be the optimal selection of treatments in patients with esophageal SCC. Nevertheless, This study has some limitations which we have re-discussed in the last paragraph of Discussion section. The main limitation was that the meta-analysis was conducted based on results reported from trials rather than individual patient data, and based on indirect comparisons but not direct comparisons. In addition, PD-1 inhibitors are likely more effective for patients with SCC and tumors with high PD-L1 expression. For those with negative or low PD-L1 level tumors, whether PD-1 inhibitors are still superior to other treatments remain uncertain. Moreover, SAEs data for Ramucirumab + CT was not provided in individual trials, and thus, we could not investigate the comparative safety profile of this treatment. All the limitations mentioned above do not allow us to reach a definitive conclusion about which was the best treatment, and our findings should be interpreted with caution.

Due to that all studies of targeted therapies did not report patients PD-L1 expression level, we could not further assess the comparative efficacy according to PD-L1 expression status. There is a need for additional phase III trials focusing on this subject.

In term of the selection of articles, all phase II and III randomized controlled trials which compared ICIs or targeted therapies with CT or BSC/placebo were included in our study, and retrospective study, phase I trials, and non-randomized trials were excluded. To achieve the network of eligible treatment comparisons, two phase III trials [28,29] which assessed CT vs BSC/placebo were also included because the control arm in several trials assessing efficacy of target agents (anlotinib, everolimus, gefitinib, and ramucirumab) was BSC/placebo. That is, the two trials actually acted as a “bridge” in this network meta-analysis. 

Fig 2 Network of eligible comparisons for the network meta-analysis. a, overall survival; b, progression-free survival. T-DXd, trastuzumab deruxtecan; CT, chemotherapy; BSC, best supportive care.

Reviewer #3: In this manuscript, the authors made a network meta-analysis about comparative efficiency of treatments for previously treated patients with advanced esophageal and esophagogastric junction cancer.

Major revision:

1. There was no quality assessment of the included studies;

2. The charts of sensitivity analysis and heterogeneity analysis were not provided.

Response: Quality assessment (S1 and S2 Fig), sensitivity analysis (S4 Table), transitivity (S4 Fig), and inconsistency (S3 Table and S5 Fig) can be found in Supporting information. Pathological type (esophageal adenocarcinoma and squamous cell carcinoma) might account for a part of heterogeneity, and we had performed the subgroup analyses accordingly. Besides, different CT regimens and schedules used in individual trials might also lead to heterogeneity, which had been discussed as a limitation in the last paragraph of Discussion section.

Reviewer #4: Based on the systematic paper review, the authors conducted a network meta-analysis to investigate which the most effective treatment for previously treated patients with advanced esophageal and esophagogastic junction cancer. They found that Ramucirumab+CT and PD-1 inhibitors were superior to CT for previously treated advanced esophageal/EGJ cancer. I think the data is valuable. They also discuss the limitation of the current study and pointing out future directions to confirm the results found in the current study.

Page 11, the first time use of network meta-analysis need to be the full name other that the abbreviation.

Response: We have revised it.

---

## [Decision Letter · Decision Letter 1]

24 May 2021

Comparative efficacy of treatments for previously treated patients with advanced esophageal and esophagogastric junction cancer: a network meta-analysis

PONE-D-20-24414R1

Dear Dr. Dang,

We’re pleased to inform you that your manuscript has been judged scientifically suitable for publication and will be formally accepted for publication once it meets all outstanding technical requirements.

Kind regards,

Ahmed Negida, MD

Academic Editor

PLOS ONE

Additional Editor Comments (optional):

Reviewers' comments:

Reviewer's Responses to Questions

**Comments to the Author**

1. If the authors have adequately addressed your comments raised in a previous round of review and you feel that this manuscript is now acceptable for publication, you may indicate that here to bypass the “Comments to the Author” section, enter your conflict of interest statement in the “Confidential to Editor” section, and submit your "Accept" recommendation.

Reviewer #3: All comments have been addressed

Reviewer #4: All comments have been addressed

2. Is the manuscript technically sound, and do the data support the conclusions?

Reviewer #3: Yes

Reviewer #4: Yes

3. Has the statistical analysis been performed appropriately and rigorously? 

Reviewer #3: Yes

Reviewer #4: I Don't Know

4. Have the authors made all data underlying the findings in their manuscript fully available?

Reviewer #3: Yes

Reviewer #4: Yes

5. Is the manuscript presented in an intelligible fashion and written in standard English?

Reviewer #3: Yes

Reviewer #4: Yes

6. Review Comments to the Author

Reviewer #3: (No Response)

Reviewer #4: This draft of the manuscript includes good solutions to the issues identified in the previous draft.

7. PLOS authors have the option to publish the peer review history of their article (what does this mean?). If published, this will include your full peer review and any attached files.

Reviewer #3: **Yes: **Wei Liu

Reviewer #4: No

---

## [Editor Report · Acceptance letter]

27 May 2021

PONE-D-20-24414R1 

Comparative efficacy of treatments for previously treated patients with advanced esophageal and esophagogastric junction cancer: a network meta-analysis 

Dear Dr. Dang:

I'm pleased to inform you that your manuscript has been deemed suitable for publication in PLOS ONE. Congratulations! Your manuscript is now with our production department. 

Kind regards, 

on behalf of

Dr. Ahmed Negida 

%CORR_ED_EDITOR_ROLE%

PLOS ONE